# Evaluating the impact of COVID-19 pandemic on complicated malaria admissions and outcomes in the paediatric Ho Teaching Hospital of the Volta Region of Ghana

Verner N. Orish[1]*, Kennedy Akake[2], Sylvester Y. Lokpo[3], Precious K. Kwadzokpui[3,4], Kokou Hefoume Amegan-Aho[5], Lennox Mac-Ankrah[1], Emily Boakye-Yiadom[1], Jamfaru Ibrahim[1], Theophilus B. Kwofie[1]

1 Department of Microbiology and Immunology, School of Medicine, University of Health and Allied Sciences, Ho, Volta Region, Ghana, 2 School of Medicine, University of Health and Allied Sciences, Ho, Volta Region, Ghana, 3 Department of Medical Laboratory Sciences, School of Allied Health Sciences, University of Health and Allied Sciences, Ho, Volta Region, Ghana, 4 Laboratory Department, Ho Teaching Hospital, Ho, Volta Region, Ghana, 5 Department of Paediatrics, School of Medicine, University of Health and Allied Sciences, Ho, Volta Region, Ghana

* vorish@uhas.edu.gh

**Data Availability Statement:** Data cannot be made publicly available for ethical reasons because it will

## Abstract

Since Ghana recorded its first cases of COVID-19 in early March 2020, healthcare delivery in the country has been hugely affected by the pandemic. Malaria continues to be an important public health problem in terms of morbidity and mortality among children, and it is responsible for significant hospital visits and admission. It is likely that, as with other illnesses, the COVID-19 pandemic may have impacted health seeking behaviour, hospital visits, and admissions of malaria among the paediatric population in Ghana. The aim of this study was to evaluate the impact of COVID-19 pandemic on the admissions and outcome of complicated malaria in the Ho Teaching Hospital of the Volta Region of Ghana. The medical records of children admitted for complicated malaria (cerebral and severe malaria) from 2016 to 2020, were obtained from the admission records of the children. Both demographics and clinical details were collected, and data was analysed using SPSS version 25 statistical software. The yearly differences in the trend and proportions of complicated malaria admissions were performed using rate comparison analysis and Pearson chi-square was used to assess the association between the various demographic factors and yearly admission rates. Clopper-Pearson test statistic was employed to determine the 95% confidence intervals of outcome variables of interest. The year 2020 had the lowest admission for complicated malaria (149, 11.5%; 95% CI: 9.7–13.5) but proportionally had, more cases of cerebral malaria (25, 16.8%; 95% CI: 10.9–24.8), and more deaths (6, 4.0%; 95% CI: 1.5–8.8), compared to the years under review. Children admitted in 2020 had the **shortest** mean stay on admission (4.34 ±2.48, p<0.001). More studies are needed to further elucidate the impact of the COVID-19 pandemic on the health of children in malaria endemic areas.

compromise patient privacy and confidentiality. These restrictions were imposed by the Research Ethics Committee of the University of Health and Allied Sciences. Contact information: Administrator, Research Ethics Committee, University of Health and Allied Sciences, PMB 31 Ho, rec@uhas.edu.gh.

**Funding:** The authors received no specific funding for this work.

**Competing interests:** The authors have declared that no competing interests exist.

## Introduction

The year 2020 saw an unprecedented rise in the global COVID-19 cases, caused by SARS--COV-2 virus, which started in Wuhan China, around November of 2019 [1]. This rise and spread of the virus to many countries in the first quarter of 2020 resulted in the WHO aptly declaring COVID-19 a pandemic on 11[th] march, 2020 [2]. Since then, the virus has infected over 270 million, killing more than 5 million people globally [3]. Morbidity and mortality from COVID-19 are not the only problems posed by the pandemic. There are significant indirect impacts such as reduced economic growth, change in social life, and healthcare delivery [4]. The COVID-19 pandemic is taking a heavy toll on healthcare delivery globally, especially in resource limited settings of developing countries, where healthcare delivery is plagued by poor infrastructure and inadequate manpower [4,5].

The COVID-19 pandemic in many resource limited areas, has overwhelmed the healthcare system, usurping the scarce manpower and resources, causing other common or endemic diseases requiring hospital care to go untreated, resulting in avoidable deaths especially in vulnerable populations [6–8]. More so, the fear caused by the pandemic and the lockdowns that followed, worsened the poor health seeking behaviour common in these settings, further resulting in delay of appropriate treatment and its dire consequences [8]. There are several common or endemic diseases requiring prompt hospital care such as tuberculosis, HIV-AIDS, pneumonia, diarrheal diseases, and malaria which can be worsened due to the COVID-19 pandemic [8,9].

Malaria remains a nagging public health issue in many developing countries causing morbidity and mortality among people living in endemic areas especially pregnant women and children [10]. Malaria caused over 270 thousand childhood deaths in 2018, constituting about 67% of all malaria deaths globally [11]. According to the WHO malaria reports of 2021, in 2020, the African continent recoded about 96% of the 627,000 global malaria deaths and children especially those under 5 years of age constituted about 80% of these deaths [12]. Malaria is one of the major reasons for hospital visits and admissions in endemic areas, and in the early days of malaria control it accounted for about 25% to 35% of all outpatient visits, 25% to 40% of inpatients and 15% to 30% of hospital mortality [13]. Although substantial progress has been made over the years in malaria control and prevention, resulting in a reduction of the hospital burden of malaria, a significant number of cases still visit the hospital with admissions and deaths [14]. In endemic areas, malaria in children caused by *Plasmodium falciparum* species, can quickly progress to complicated malaria such as **cerebra**l malaria (CM) and other forms of severe malaria such as severe malaria anaemia (SMA), severe malaria with hypoglycaemia, and severe malaria with renal complications [15]. Complicated malaria can be fatal if prompt hospital admission with appropriate treatment does not take place [15]. Factors impeding prompt access to healthcare when children develop symptoms of malaria or decreasing the alacrity of parents and care givers to seek care, might reduce hospital visits with fatal consequences [16]. The COVID-19 pandemic with the lockdowns and fears, might have created an additional barrier or worsened already existing barriers that are replete in malaria endemic areas [5,8].

Ghana reported its first 2 cases of COVID-19 on 12 of March, 2020 and within 6 months all regions in the country have reported cases of COVID-19 cases, with the Volta Region reporting first 9 cases on 12 of April, 2020 [17,18]. Although there was no nationwide lockdown, some parts of the capital, Accra and other initial hotspots for the pandemic were on lockdown, coupled with complete closure of all school, land borders as well as suspension of international air travels [17]. In addition to this, people were urged to stay indoors and restrict movements, avoid crowded places, practice social distancing, use a face mask, and practice hand hygiene

with hand washing or hand sanitizers. With a total of over 135,000 cases, and slightly over 1000 deaths, as at December 2021 [19], many in the country were indeed afraid of contracting the virus, which might have reduced their health seeking behaviour [17]. Malaria among children is still a significant public health issue in Ghana and one of the common reasons for hospital visits and admission [20,21]. With reports suggesting that malaria contributes significantly to hospital visits and admissions [21], it is likely that the COVID-19 pandemic and the subsequent restriction of movements could have decrease access to healthcare or delayed seeking care among children with malaria. In fact, the mere fear of contracting COVID-19, especially during the peak period of the COVID-19 pandemic, where the hospital was perceived by some parents as an infection hub, could significantly have contributed to high hesitancy in seeking appropriate healthcare for children with malaria. Thus, this study evaluated the impact of the pandemic on the admissions and outcomes of complicated malaria among children at the Ho Teaching Hospital in the Volta Region of Ghana.

## Methodology

### Study site

This study was conducted in the Paediatric Department of the Ho Teaching Hospital. The facility is the major tertiary referral hospital in the region. The hospital also attracts patients from other regions in the country and from neighbouring countries of the West African sub-region. It has an overall 354 bed capacity with major specialty and subspecialty units in Medicine, Surgery, Obstetrics and Gynaecology and Paediatrics. The paediatric Department consists of an outpatient department and 3 wards namely, Neonatal Intensive Care Unit (NICU), Babies Unit (children aged 0 to 6 months), Children's Unit (children aged 6 months to 12 years). The children's ward is a 25-bed unit, with an average yearly admission of about 950 children.The Ho Teaching Hospital is located in the Ho municipality, the capital of the Volta Region. The municipality is located between latitudes 6˚20" N and 6˚55" N and longitudes 0˚12" E and 0˚53" E, occupying 11.5% of the region's total land area. The municipality is made up of mountainous and lowland areas. The mean annual temperature ranges from 16.5˚C to 37.8˚C. There are two rainy seasons, a major one between March and July, and a minor one between August and November. The mean annual rainfall is between 20.1mm and 192mm. The vegetative zones are a moist deciduous forest in the hilly areas and savannah woodlands in the rest of the municipality.

### Study design and procedure

This was a descriptive retrospective study of clinical records of children admitted to the children's ward of the Ho Teaching Hospital in the time period spanning January, 2016 to December, 2020. Data were retrieved from the ward's admission and discharge records. All records of children admitted for complicated malaria within the study period, were collected. Demographic information like age, sex, religion, location of residence and mode of payment of hospital bills were obtained. Clinical information such as diagnosis of severe malaria anaemia and cerebral malaria, comorbidities, duration of admission, outcome of admission (discharge, referral or death), were also obtained.

### Data analysis

All information extracted from the admission register was entered into Microsoft excel 2016 spread sheet for data verification and cleaning and subsequently exported to the Statistical package for the Social Sciences (SPSS) version 22, where analysis was performed. Frequency

distributions as well as proportions were computed for sociodemographic and clinical characteristics of the admitted children. Analysis of the yearly differences in the trend and proportions of complicated malaria admissions was performed using the Pearson-chi square test. The Clopper-Pearson test statistic was used to determine the 95% confidence intervals of the outcome variables of interest. Results were presented as proportion with 95% confidence intervals and findings with p-value less than 5% were considered statistically significant.

### Ethical considerations

Ethical approval for this study was obtained from the Ethics Committee of the University of Health and Allied Sciences (UHAS-REC A. 12 [8 9] 20–21). Permission was obtained from hospital management to access the hospital records. All records of the children for this retrospective study were collected under strict anonymity. Informed consent requirement for study was waived by the Ethics Committee of the University of Health and Allied Sciences.

## Results

A total of 1,296 children were admitted for complicated malaria from January 2016 to December 2020. Of these, 673(51.93%) were males while 623(48.07%) were females. The majority of the children were between the ages of 1–4 years (775, 59.81%), belonged to the Christian faith (1110, 85.65%), were enrolled into the National Health Insurance Scheme [NHIS] (1114, 85.96%), and majority resided outside of the municipality (662, 51.1%) (Table 1).

Table 1 also shows the frequency distribution of the clinical characteristics of the admitted children. More children were admitted for **severe malaria** (SM) (1183, 91.28%) than for **cerebral malaria** (1183, 8.72%). The majority of the children, had no comorbidities (702, 54.17%), were discharged (1263, 97.45%), and majority spent 2–7 days on admission (1083, 83.56%). A total of 27(2.08%) children died within the study period and majority of these deaths occurred within first 24 hrs. on admission (21, 1.62%), with cerebral malaria marginally causing more deaths (14, 1.02%) than Several Malaria (13, 1.00%).

Table 2 shows the clinical and demographic characteristics of the admitted children stratified by the years under review. There were no significant differences between the years under review for gender (p = 0.21), age distribution (0.12), mean age (0.18), and residence (0.52). However, there was significant difference in the mean duration of stay on admission between the years under review with children admitted in year 2020 (4.34 ±2.48, p<0.001), staying the shortest on admission compared to other years. Generally, there was a significant difference in the change in admission rate between the year 2020 and the other years under review where the difference between the year 2016 and 2020 was 6.6%(95% CI:3.6–9.5; p<0.001) but rises to as high as 13.4%(95% CI:10.1–16.6; p<0.001) between 2019 and 2020. Proportionally, there was significantly more children admitted with cerebral malaria in the year 2020 (16.8%, (11.2–23.8)) compared 2016 (12.4%, (8.5–17.3)), 2017 (8.5%, (5.5–12.5)), 2018 (4.7%, (2.6–7.6)) and 2019 (6.5%, (4.1–9.8)). Again, there was significantly more deaths among the children admitted in 2020 (4.0% (1.5–8.6)) compared to the other years under review and the only year with children referred to intensive care unit or referred out of the hospital.

Fig 1 shows the gender and age distribution of the 27 children who died on admission in the study period. There were 15(55.56%) females and 12(44.44) males. Nineteen (70.37%) children were under 5 years old and 8 (29.62%) above 5 years. There was no significant difference between the years under review (p = 0.68), although, 2020 recoded more children above 5 years that died on admission.

Table 3 shows the monthly trend of malaria admissions of the 5 years under review. Generally, the year 2020 had the lowest monthly admission rate compared to the other years under

**Table 1. Demographics and clinical characteristics of children admitted with complicated malaria.**

| Characteristic | Frequency | Percentage |
|---|---|---|
| **Sex** | | |
| Female | 623 | 48.07 |
| Male | 673 | 51.93 |
| **Age** | | |
| <1 year | 93 | 7.17 |
| 1–4 years | 775 | 59.81 |
| >5 years | 428 | 33.02 |
| **Religion** | | |
| Christian | 1110 | 85.65 |
| Muslim | 186 | 14.35 |
| **Residence** | | |
| Within Ho Municipality | 634 | 48.9 |
| Outside the municipality | 662 | 51.1 |
| **Mode of payment of hospital bills** | | |
| Public NHIS | 1114 | 85.96 |
| Private Cash | 182 | 14.04 |
| **Complicated malaria** | | |
| Cerebral Malaria (CM) | 113 | 8.72 |
| Severe Malaria Anaemia (SMA) | 1183 | 91.28 |
| **Comorbidities,** | | |
| Present | 594 | 45.83 |
| Absent | 702 | 54.17 |
| **Duration of Stay on admission** | | |
| 1 day | 20 | 1.54 |
| 2–7 days | 1083 | 83.56 |
| 8–14 days | 136 | 10.5 |
| 15–29 days | 45 | 3.47 |
| 30 days/+ | 12 | 0.93 |
| **Outcomes** | | |
| Recovery/Discharge | 1263 | 97.45 |
| Referral | 6 | 0.47 |
| Mortality | 27 | 2.08 |
| **Death with 24 hrs** | 21 | 1.62 |
| **Deaths caused by CM** | 14 | 1.08 |
| **Deaths caused by SMA** | 13 | 1.00 |

review with a consistent lowest rate of admission documented from the month March down to December.

Fig 2 shows the seasonal trend for malaria admissions for the 5 years under review and for 2020. The lowest cumulative number of admissions was noted in the month March for the 5 years under review. There was a steady increase in admissions and it peaked in July. The year 2020 had an almost similar pattern, with lowest admission noted in March, April and September.

Fig 3 compared the yearly malaria admission with the total admission in paediatric unit over the 5 years under review. There was total of 4920 admissions in the paediatric ward over the 5 years period with malaria admissions representing 26.3% (1296) of all admissions. The year 2020 also significantly had the lowest proportion of total admissions compared to the other years (894, 18.2%; (17.0–19.3)).

**Table 2. Characteristics of children admitted with complicated malaria stratified by years under review.**

| Characteristics | 2016 n[%(95% CI] | 2017 n[%(95% CI] | 2018 n[%(95% CI] | 2019 n[%(95% CI] | 2020 n[%(95% CI] | p |
|---|---|---|---|---|---|---|
| **Total** | 234[18.1(16.0–20.3)] | 271[20.9(18.7–23.2)] | 320[24.7(22.4–27.1)] | 322[24.8(22.5–27.3)] | 149[11.5(9.8–13.4)] | |
| Δ Admission rate (%) | 6.6(3.6–9.5)* | 9.4(6.3–12.5)** | 13.2(9.9–16.5)*** | 13.4(10.1–16.6)**** | | |
| P | <0.001 | <0.001 | <0.001 | <0.001 | | |
| **Sex** | | | | | | |
| Female | 121[51.7(45.1–58.3)] | 124[45.8(39.7–51.9)] | 156[48.8(43.2–54.4)] | 142[44.1(38.6–49.7)] | 80[53.7(45.3–61.9)] | 0.210 |
| Male | 113[48.3(41.7–54.9)] | 147[54.2(48.1–60.3)] | 164[51.3(45.6–56.8)] | 180[55.9(50.3–61.4)] | 69[46.3(38.1–54.7)] | |
| **Age** | | | | | | |
| <1 year | 14[6.0(3.3–9.8)] | 25[9.2(6.1–13.3)] | 26[8.1(5.4–11.7)] | 19[5.9(3.6–9.1)] | 9[6.0(2.8–11.2)] | 0.124 |
| 1–4 years | 145[62.0(55.4–68.2)] | 173[63.8(57.8–69.6)] | 176[55.0(49.4–60.5)] | 193[59.9(54.4–65.3)] | 88[59.1(50.7–67.0)] | |
| >5 years | 75[32.1(26.1–38.4)] | 73[26.9(21.7–32.6)] | 118[36.9(31.6–42.4)] | 110[34.2(29.0–39.6)] | 52[34.9(27.3–43.1)] | |
| Age (years.) Mean ± SD | 3.69±2.86 | 3.46±2.78 | 3.88±2.77 | 3.94±2.79 | 3.99±2.51 | 0.181 |
| **Residence** | | | | | | |
| Within Ho Municipality | 111[47.4(40.9–54.0)] | 134[49.5(43.3–55.6)] | 168[52.5(46.9–58.1)] | 155[48.1(42.6–53.7)] | 66[44.3(36.2–52.7)] | 0.520 |
| Outside the municipality | 123[52.6(46.0–59.1)] | 137[50.6(44.4–56.7)] | 152[47.5(41.9–53.1)] | 167[51.9(46.3–57.4)] | 83[55.7(47.3–63.8)] | |
| **Payment of hospital bills** | | | | | | |
| Public NHIS | 186[79.5(73.7–84.5)] | 236[87.1(82.5–90.8)] | 277[86.6(82.3–90.1)] | 284[88.2(84.2–91.5)] | 131[87.9(81.6–92.7)] | 0.050 |
| Private Cash | 48[20.5(15.5–26.3)] | 35[12.9(9.2–17.5)] | 43[13.4(9.9–17.7)] | 38[11.8(8.5–15.8)] | 18[12.8(7.3–18.4)] | |
| **Final Diagnosis** | | | | | | 0.002 |
| Cerebral Malaria | 29[12.4(8.5–17.3)] | 23[8.5(5.5–12.5)] | 15[4.7(2.6–7.6)] | 21[6.5(4.1–9.8)] | 25[16.8(11.2–23.8)] | |
| Severe Malaria | 205[87.6(82.7–91.5)] | 248[91.5(87.5–94.5)] | 305[95.3(92.4–97.4)] | 301[93.5(90.2–95.9)] | 124[83.2(76.2–88.8)] | |
| **Comorbidities** | | | | | | <0.001 |
| Present | 111[47.4(40.9–54.0)] | 160[59.0(52.9–65.0)] | 136[42.5(37.0–48.1)] | 137[42.6(37.1–48.1)] | 50[33.6(26.0–41.7)] | |
| Absent | 123[52.6(46.0–59.1)] | 111[41.0(35.0–47.1)] | 184[57.5(51.9–63.0)] | 185[57.5(51.9–62.9)] | 99[66.4(58.3–74.0)] | |
| **Length of stay on admission** | | | | | | |
| 1 day | 3[1.3(0.3–3.7)] | 2[0.7(0.1–2.6)] | 7[2.2(0.9–4.5)] | 4[1.2(0.3–3.1)] | 4[2.7(0.7–6.7)] | 0.090 |
| 2–7 days | 193[82.5(77.0–87.1)] | 211[77.9(72.4–82.7)] | 274[85.6(81.3–89.3)] | 277[86.0(81.8–89.6)] | 128[85.9(79.3–91.1)] | |
| 8–14 days | 25[10.7(7.0–15.4)] | 39[14.4(10.4–19.1)] | 32[10.0(6.9–13.8)] | 28[8.7(5.9–12.3)] | 12[8.1(4.2–13.6)] | |
| 15–29 days | 10[4.3(2.1–7.7)] | 17[6.3(3.7–9.9)] | 6[1.9(0.7–4.0)] | 9[2.8(1.3–5.2)] | 3[2.0(0.4–5.8)] | |
| 30 days/+ | 3[1.3(0.3–3.7)] | 2[0.7(0.1–2.6)] | 1[0.3(0.0–1.7)] | 4[1.2(0.3–3.1)] | 2[1.3(0.2–4.8)] | |
| Mean ± SD(days) | 5.76 ±6.03 | 6.65 ±6.22 | 5.01 ±4.94 | 5.20 ±5.13 | 4.34 ±2.48 | <0.001 |
| **Outcomes** | | | | | | |
| Recovery/Discharge | 228[97.4(94.5–99.1)] | 263[97.0(94.3–98.7)] | 318[99.4(97.8–99.9)] | 317[98.5(96.4–99.5)] | 137[92.0(86.4–95.8)] | <0.001 |
| Referrals | 0[0.00] | 0[0.00] | 0[0.0] | 0[0.00] | 6[4.0(1.5–8.6)] | |
| Mortality | 6[2.6(0.9–5.5)] | 8[3.0(1.3–5.7)] | 2[0.6(0.1–2.2)] | 5[1.6(0.5–3.6)] | 6[4.0(1.5–8.6)] | |
| **Time of death** | | | | | | |
| Within 24 hours | 5[83.3(35.9–99.6)] | 5[62.5(24.5–91.5)] | 2[100.0(15.8–100)] | 4[80.0(28.4–99.5)] | 5[83.3(35.9–99.6)] | 0.750 |
| After 24 hours | 1[16.7(0.4–64.1)] | 3[37.5(8.5–75.5)] | 0[0.00] | 1[20.0(0.5–71.6)] | 1[16.7(0.4–64.1)] | |

NHIS-National Health Insurance Scheme; Δ - Change in admission rate comparing

* 2016 to 2020

** 2017 to 2020

*** 2018 to 2020

**** 2019 to 2020.

## Discussion

Though the negative impact of the COVID-19 pandemic will probably linger for many years to come, 2020 might be the year with the worst impact of the pandemic [22]. The WHO

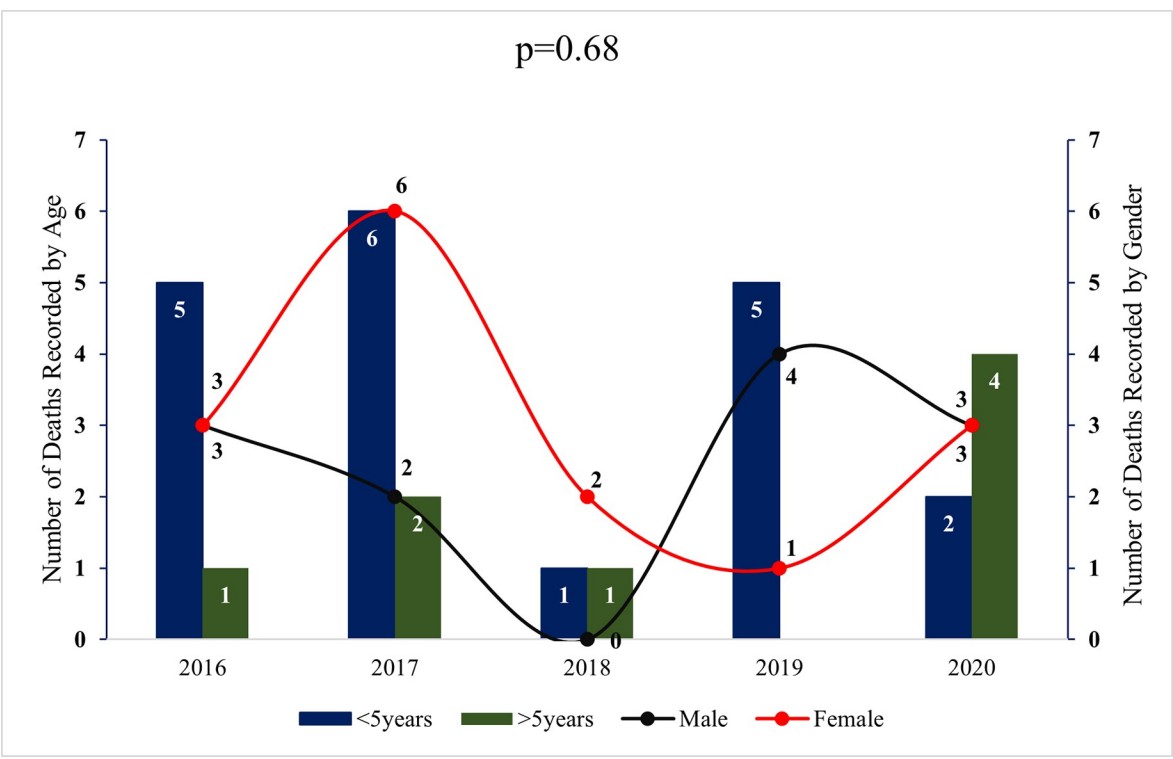

**Fig 1. Mortality, age and gender distribution for the 5 years under review.**

malaria report for 2020, showing an excess of malaria deaths compared to 2018 and 2019, was linked to the negative impact of the COVID-19 pandemic [23,24]. In Ghana, the pandemic most likely affected health seeking behaviour for various endemic illnesses especially malaria, but the extent of the negative impact is difficult to ascertain since there are no data to verify [17].

This study showed that of all the 1296 admitted cases of complicated malaria over the study period, the year 2020 recorded the lowest proportion (11.5%; 95% CI: (9.8–13.4)) of admissions. Compared to other years, 2020 recorded the lowest monthly figures for malaria admissions starting from the month of March to December. It is very likely that this finding was as

**Table 3. Month-to-month trend of malaria admission in the paediatric unit from 2016 to 2020.**

| Month | | Jan | Feb | Mar | April | May | Jun | Jul | Aug | Sept | Oct | Nov | Dec |
|---|---|---|---|---|---|---|---|---|---|---|---|---|---|
| **2016** | Frequency | 6[7.8] | 7[13.5] | 6[18.8] | 15[21.7] | 23[18.9] | 28[19.2] | 37[23.1] | 30[19.0] | 31[19.9] | 18[16.8] | 15[12.6] | 18[18.4] |
| | 95% CI | 2.9–16.2 | 5.6–25.8 | 7.2–36.4 | 12.7–33.3 | 12.3–26.9 | 13.1–26.5 | 16.8–30.4 | 13.2–26.0 | 13.9–27.0 | 10.3–25.3 | 7.2–19.9 | 11.3–27.5 |
| **2017** | Frequency | 9[11.7] | 15[28.9] | 10[31.3] | 17[24.6] | 29[23.8] | 50[34.3] | 30[18.8] | 34[21.5] | 31[19.9] | 19[17.8] | 14[11.8] | 13[13.3] |
| | 95% CI | 5.5–21.0 | 17.1–43.1 | 16.1–50.0 | 15.1–36.5 | 16.5–32.3 | 26.6–42.5 | 13.0–25.7 | 15.4–28.8 | 13.9–27.0 | 11.0–26.3 | 6.6–19.0 | 7.3–21.6 |
| **2018** | Frequency | 18[23.4] | 8[15.4] | 9[28.1] | 19[27.5] | 33[27.1] | 19[13.0] | 42[26.3] | 33[20.9] | 31[19.9] | 34[31.8] | 44[37.0] | 30[30.6] |
| | 95% CI | 14.5–34.4 | 6.9–28.1 | 13.7–46.7 | 17.5–39.6 | 19.4–35.8 | 8.0–19.6 | 19.6–33.8 | 14.8–28.1 | 13.9–27.0 | 23.1–41.5 | 28.3–46.3 | 21.7–40.7 |
| **2019** | Frequency | 25[32.5] | 13[25.0] | 2[6.3] | 14[20.3] | 22[18.0] | 32[21.9] | 38[23.8] | 44[27.9] | 56[35.9] | 22[20.6] | 29[24.4] | 25[25.5] |
| | 95% CI | 22.2–44.1 | 14.0–38.9 | 0.8–20.8 | 11.6–31.7 | 11.7–26.0 | 15.5–38.5 | 17.4–31.1 | 21.0–35.5 | 28.4–44.0 | 13.4–29.5 | 17.0–33.1 | 17.2–35.3 |
| **2020** | Frequency | 19[24.7] | 9[17.3] | 5[15.6] | 4[5.8] | 15[12.3] | 17[11.64] | 13[8.1] | 17[10.8] | 7[4.5] | 14[13.1] | 17[14.3] | 12[12.3] |
| | 95% CI | 15.6–35.8 | 8.2–30.3 | 5.3–32.8 | 1.6–14.2 | 7.0–19.5 | 6.9–18.0 | 4.4–13.5 | 6.4–16.7 | 1.8–9.0 | 7.3–21.0 | 8.5–21.9 | 6.5–20.4 |
| **Total** | | 77[100] | 52[100] | 32[100] | 69[100] | 122[100] | 146[100] | 160[100] | 158[100] | 156[100] | 107[100] | 119[100] | 98[100] |

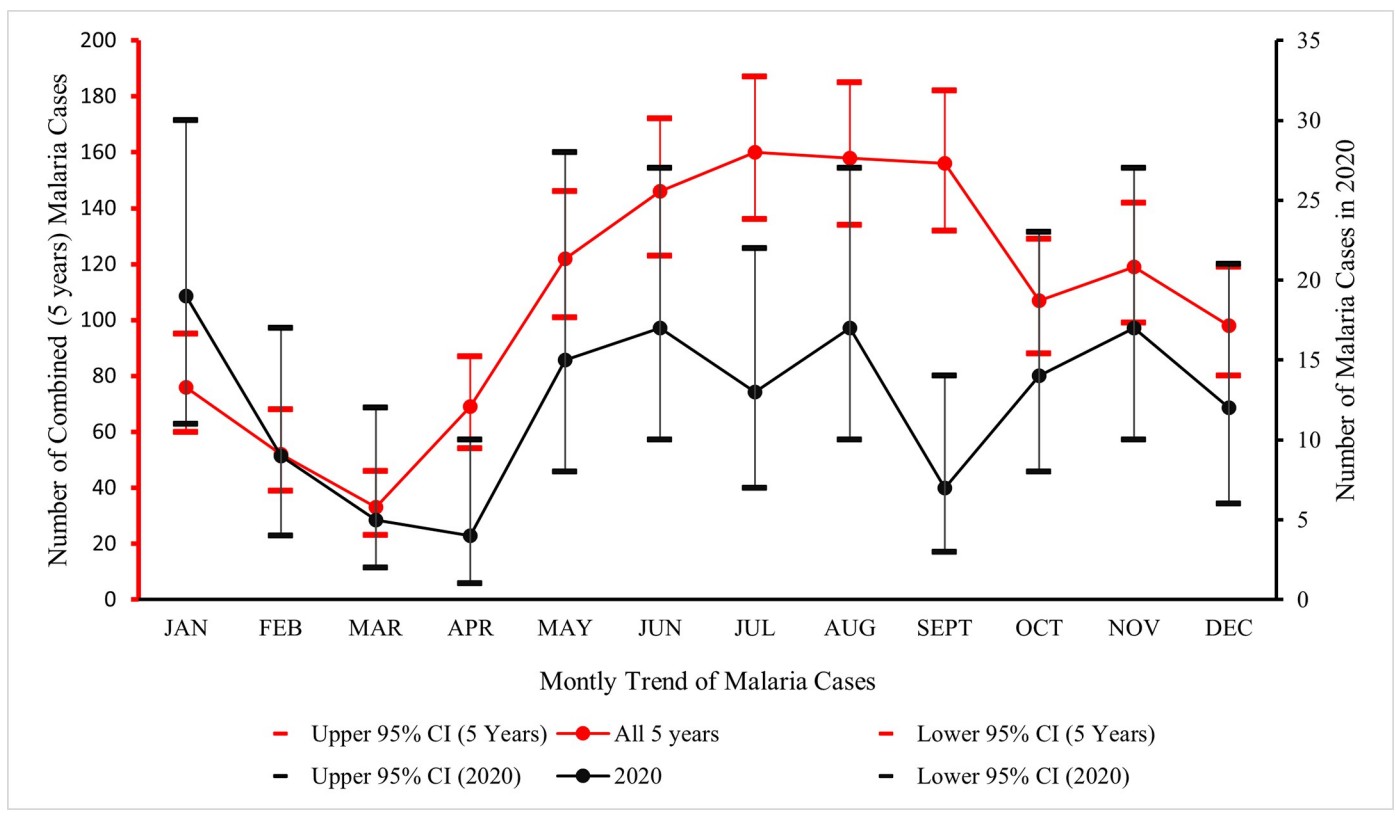

**Fig 2. Seasonal trend of complicated malaria admission in the children ward.**

result of the reduced health seeking behaviour caused by fear of the pandemic-which started in early March in Ghana, causing reduced hospital visits and subsequent admission rates [5,9,17]. However, it is important to note that despite the decline in malaria admission in 2020, the trend almost mimicked the typical seasonal pattern, with a slight increase in admissions in the rainy season (April to July) and a dip in the dry season (August), and an increase in the second rainy season (September to October) followed by a drop in the next cycle of the dry season [25,26].

Severe malaria (91%), unlike cerebral malaria (9%) was the most prevalent complicated malaria admitted during the study period. This finding is in agreement with some studies done in Ghana [27,28], though some studies conducted in other parts of Africa, yielded contrasting results [29–31]. Of the 149 children admitted for complicated malaria in 2020, there were proportionally more cerebral malaria cases (17%) compared to the other years under review. With studies suggesting that cerebral malaria is a more potent predictor of death than other forms of complicated malaria, [16,29,31], it might explain why 2020 had a higher percentage of mortality and lower percentage recovery compared to the other years. The year 2020 was the only year in the study period when some admitted children were referred to either the intensive care unit of the hospital or referred out to the Korle Bu Teaching Hospital in Accra. The outcome of these referred children were not known in this study, but it seem likely the bad prognosis that initiated referral at the first place, especially among children with cerebral malaria were highly life threatening and often result in death [32].

The majority of the children spent between 2–7 days on admission for the overall study period. Children admitted with complicated malaria in the year 2020, significantly spent the

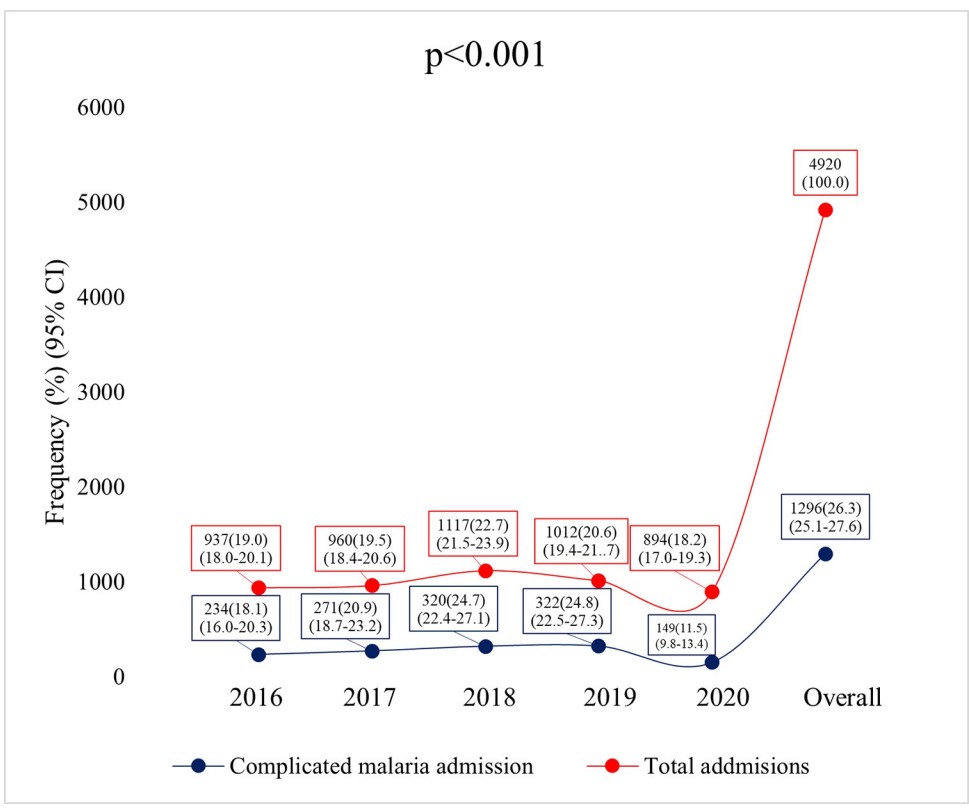

**Fig 3. Trend of total admission compared to complicated malaria admission.**

shortest average length of stay on admission (4.34 ±2.48). This, further buttress the point that 2020 had children with the severest forms of complicated malaria on admission, as reduced length of stay on admission has been linked with severe illness which is usually followed by death in the first 24 hours [33]. Again, the reason for this might be a delay in seeking treatment, a common scenario in malaria endemic areas [16,33] and probably worsened by the fear of the pandemic [17]. Another possible factor that might have been responsible for the relatively short length of stay on admission for 2020, is the practice of preventing unnecessary prolonged stay in hospital in order to prevent spread of the COVID-19 among inpatients. This could have influenced clinicians to discharge the children as soon as their condition began to improve. This practice was heavily encouraged in the early days of the pandemic and in some settings became an approved guideline for rapid discharge of patients who are clinically ready [34,35].

This study showed that the majority of the children admitted were under 5 years of age and this was consistent for all the years under study, including 2020. This is typical of the malaria morbidity among children in malaria endemic areas, with areas of high malaria transmission (like the study area) affecting younger children and low transmission areas affecting older children [36–38]. Also, there is corresponding high mortality among children in areas of high malaria transmission as seen in this study [37,38]. This finding was again consistent for each year under study except 2020, where death was recorded more for older children, although this finding was not significant.

The pandemic did not only affect the admission for complicated malaria as other admission in the children's ward were also affected. Total admission for the year 2020 was significantly

the lowest for the study period. This probably suggests that many other childhood illnesses, such as diarrheal diseases, bacterial infections and others, would have lacked proper treatment in health facilities, during this period, resulting in dire consequences [17,39].

The limitation noted in this study was the inability to review in detail the clinical data of these children admitted with complicated malaria to clearly determine the severity of their clinical condition. However, with the basic data obtained, this study provided interesting findings in an attempt to answer the question of the impact of the pandemic on malaria admissions and outcomes.

## Conclusion

The year 2020 saw a lower rate of admission of children with complicated malaria with more children admitted for cerebral malaria compared to the other years where severe malaria was more common. Children admitted in 2020 had the shortest length of stay on admission and there were relatively more deaths compared to the other years under review. The impact of the pandemic on admissions and the overall health of children living in endemic areas may not be fully understood. However, more studies like this are needed to help shed more light on the impact of the ongoing pandemic so that lessons learned can help mitigate the effect of the present and any future pandemic.

## Acknowledgments

We expressed our profound gratitude to the staff of the paediatric and records units for their support in making this work a success.

## Author Contributions

**Conceptualization:** Verner N. Orish.

**Data curation:** Verner N. Orish, Kennedy Akake, Precious K. Kwadzokpui.

**Formal analysis:** Verner N. Orish, Kennedy Akake, Precious K. Kwadzokpui.

**Investigation:** Verner N. Orish.

**Methodology:** Verner N. Orish, Kennedy Akake, Sylvester Y. Lokpo, Precious K. Kwadzokpui.

**Project administration:** Verner N. Orish.

**Resources:** Verner N. Orish.

**Software:** Verner N. Orish.

**Supervision:** Verner N. Orish.

**Validation:** Verner N. Orish, Sylvester Y. Lokpo.

**Visualization:** Verner N. Orish, Precious K. Kwadzokpui.

**Writing – original draft:** Verner N. Orish.

**Writing – review & editing:** Verner N. Orish, Kennedy Akake, Sylvester Y. Lokpo, Precious K. Kwadzokpui, Kokou Hefoume Amegan-Aho, Lennox Mac-Ankrah, Emily Boakye-Yiadom, Jamfaru Ibrahim, Theophilus B. Kwofie.

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
