## [Decision Letter · Decision Letter 0]

7 Apr 2022

PGPH-D-22-00146

Evaluating the impact of COVID-19 pandemic on complicated malaria admissions and outcomes in the paediatric Unit of the Ho teaching hospital in the Volta region of Ghana

Dear Dr. Verner Ndudri Orish

Thank you for submitting your manuscript to PLOS Global Public Health. After careful consideration, we feel that it has merit but does not fully meet PLOS Global Public Health’s publication criteria as it currently stands. Therefore, we invite you to submit a revised version of the manuscript that addresses the points raised during the review process.

We look forward to receiving your revised manuscript.

Kind regards,

Srinivasa Rao Mutheneni, PhD

Academic Editor

Journal Requirements:

1. You indicated that you had ethical approval for your study. In your Methods section, please ensure you have also stated whether you obtained consent from parents or guardians of the minors included in the study or whether the research ethics committee or IRB specifically waived the need for their consent.

2. Please amend your Financial Disclosure statement. If you did not receive any funding for this study, please simply state: “The authors received no specific funding for this work.”

3. Please update your Competing Interests statement. If you have no competing interests to declare, please state: “The authors have declared that no competing interests exist.”

4. In the online submission form, you indicated that “DATA ARE AVAILABLE FROM THE HOSPITAL RECORD AND WILL BE PROVIDED ON REQUEST”. All PLOS journals now require all data underlying the findings described in their manuscript to be freely available to other researchers, either 1. In a public repository, 2. Within the manuscript itself, or 3. Uploaded as supplementary information.

5. Please provide separate figure files in .tif or .eps format only and remove any figures embedded in your manuscript file. Please ensure that all files are under our size limit of 20MB.

Additional Editor Comments (if provided):

1. Reference are not codded properly. E.g. see reference-19 coated in the text page no-4.

2. Found capital letters in Abruptly in the document. e.g. page-3 Severe malaria in 2nd paragraph; page-4 Outpatient, Department.

3. Update the disease burden with latest reports.

Reviewers' comments:

Reviewer's Responses to Questions

**Comments to the Author**

1. Does this manuscript meet PLOS Global Public Health’s publication criteria? Is the manuscript technically sound, and do the data support the conclusions? The manuscript must describe methodologically and ethically rigorous research with conclusions that are appropriately drawn based on the data presented.

Reviewer #1: Yes

Reviewer #2: Partly

2. Has the statistical analysis been performed appropriately and rigorously?

Reviewer #1: Yes

Reviewer #2: No

3. Have the authors made all data underlying the findings in their manuscript fully available (please refer to the Data Availability Statement at the start of the manuscript PDF file)?

Reviewer #1: Yes

Reviewer #2: Yes

4. Is the manuscript presented in an intelligible fashion and written in standard English?

Reviewer #1: Yes

Reviewer #2: Yes

5. Review Comments to the Author

Reviewer #1: The study provides valuable information on the impact of the COVID19 pandemic among the children affected with complicated malaria. However, the following points should be addressed to make it more informative and understanding.

1. Has covid 19 impacted the approach of patients to admit to the hospital?

2. Page no. 6: “Malaria caused over 270 thousand childhood deaths in 2018, constituting about 67% of all malaria deaths globally” [11]:-In the interest of readers, WHO Malaria report 2021 should also be cited.

3. Page no. 7 “With malaria among children still a significant public health issue in Ghana19, no study has looked at the impact of the pandemic on this.” The sentence should be revised.

4. Page no. 11: In table 1, CM and SMA should be described as Cerebral malaria (CM) and severe malaria anemia (SMA).

Optionally, the information on the status of malaria in the country and national treatment guidelines for the treatment of children should also be addressed.

Reviewer #2: The manuscript is interesting and well written. However it is unclear how the pandemic may have changed the malaria burden among children. Perhaps access to healthcare, but unlikely the true burden (prevalence). The authors should clarify the true objectives and distinguish between what they are measuring and the underlying mechanism. I have several comments that should be addressed:

Typo on the last sentence on page 3 - the sentence ends with ‘the COVID-19’ where it should read ‘the COVID-19 pandemic’.

In the intro page 3, it is unclear why the authors cite hospital visits from prior to control programs, when the baseline of hospital visits for this study will come from the era of malaria control programs. These are now outdated statistics

Specie should be species.

Cerebral and Severe do not need to be capitalized

The COVID pandemic may have created an additional barrier to healthcare seeking, but it certainly is not the only barrier and did not create these barriers.

Last paragraph page 3 - first 2 cases of COVID, it just says first 2 cases without specifying cases of what.

The authors should define what measures were included in the lock down, this is a fairly generic term.

For the yearly differences, using a Pearson Chi2 test will test for differences between all of the years. Therefore the results don’t match with the analysis. For example, the authors state that there were significantly more children admitted with cerebral malaria in 2020 compared to other year. The chi2 test would not show this, just that there were differences between the years, not that 2020 was statistically significantly higher than all the others. This would be better shown and interpretable by having confidence intervals in table 2 to see if 2020 was different from the other years. As it is, all the table tells is that the years are different from each other.

The graphs should also have confidence bands or intervals to show the statistical significance, rather than just p values.

6. PLOS authors have the option to publish the peer review history of their article (what does this mean?). If published, this will include your full peer review and any attached files.

**Do you want your identity to be public for this peer review?** For information about this choice, including consent withdrawal, please see our Privacy Policy.

Reviewer #1: **Yes: **Kuldeep Singh

Reviewer #2: No

---

## [Decision Letter · Decision Letter 1]

10 Jun 2022

PGPH-D-22-00146R1

Evaluating the impact of COVID-19 pandemic on complicated malaria admissions and outcomes in the paediatric Unit of the Ho teaching hospital in the Volta region of Ghana

Dear Dr. verner Ndudiri orish,

Thank you for submitting your manuscript to PLOS Global Public Health. After careful consideration, we feel that it has merit but does not fully meet PLOS Global Public Health’s publication criteria as it currently stands. Therefore, we invite you to submit a revised version of the manuscript that addresses the points raised during the review process.

We look forward to receiving your revised manuscript.

Kind regards,

Srinivasa Rao Mutheneni, PhD

Academic Editor

Journal Requirements:

Additional Editor Comments (if provided):

Reviewers' comments:

Reviewer's Responses to Questions

**Comments to the Author**

1. If the authors have adequately addressed your comments raised in a previous round of review and you feel that this manuscript is now acceptable for publication, you may indicate that here to bypass the “Comments to the Author” section, enter your conflict of interest statement in the “Confidential to Editor” section, and submit your "Accept" recommendation.

Reviewer #1: All comments have been addressed

Reviewer #2: (No Response)

2. Does this manuscript meet PLOS Global Public Health’s publication criteria? Is the manuscript technically sound, and do the data support the conclusions? The manuscript must describe methodologically and ethically rigorous research with conclusions that are appropriately drawn based on the data presented.

Reviewer #1: Yes

Reviewer #2: Yes

3. Has the statistical analysis been performed appropriately and rigorously?

Reviewer #1: Yes

Reviewer #2: Yes

4. Have the authors made all data underlying the findings in their manuscript fully available (please refer to the Data Availability Statement at the start of the manuscript PDF file)?

Reviewer #1: Yes

Reviewer #2: Yes

5. Is the manuscript presented in an intelligible fashion and written in standard English?

Reviewer #1: Yes

Reviewer #2: Yes

6. Review Comments to the Author

Reviewer #1: The authors have addressed all the issues and seems interesting to the scientific community.

Reviewer #2: Most of the comments from the previous review have been addressed. However there are still some minor revisions that should be addressed. There are some typos and errors in language that should be corrected before publication. Those are:

Abstract: Page 2 - ‘Children admitted in 2020 had the shorted mean stay on admission’. Should be corrected to shortest.

Severe Malaria and Cerebral Malaria are still capitalized throughout when they shouldn’t be.

Should read ‘The majority of children’ instead of ‘Majority of children’

There is an extra parentheses in the third paragraph of the discussion.

In the fourth paragraph of the discussion it should read ‘The majority of children’ rather than ‘majority of children’.

Additionally:

Introduction: first paragraph. When citing the COVID statistics of cases and deaths it would be helpful to reference the time period, such as , ‘as of June 2022’ since these are going up every day.

Suggest editing the last sentence of page 4 to ‘might have increased these barriers’. The pandemic didn’t create all the barriers to healthcare and health seeking as the authors describe earlier in the paragraph. Many barriers exist, but the pandemic didn’t create all of these barriers, but added substantially to them.

7. PLOS authors have the option to publish the peer review history of their article (what does this mean?). If published, this will include your full peer review and any attached files.

**Do you want your identity to be public for this peer review?** For information about this choice, including consent withdrawal, please see our Privacy Policy.

Reviewer #1: No

Reviewer #2: No

---

## [Decision Letter · Decision Letter 2]

19 Aug 2022

Evaluating the impact of COVID-19 pandemic on complicated malaria admissions and outcomes in the paediatric Unit of the Ho teaching hospital in the Volta region of Ghana

PGPH-D-22-00146R2

Dear Verner Ndudiri Orish,

We are pleased to inform you that your manuscript 'Evaluating the impact of COVID-19 pandemic on complicated malaria admissions and outcomes in the paediatric Unit of the Ho teaching hospital in the Volta region of Ghana' has been provisionally accepted for publication in PLOS Global Public Health.

Best regards,

Srinivasa Rao Mutheneni, PhD

Academic Editor

Reviewer Comments (if any, and for reference):

Reviewer's Responses to Questions

**Comments to the Author**

1. If the authors have adequately addressed your comments raised in a previous round of review and you feel that this manuscript is now acceptable for publication, you may indicate that here to bypass the “Comments to the Author” section, enter your conflict of interest statement in the “Confidential to Editor” section, and submit your "Accept" recommendation.

Reviewer #1: All comments have been addressed

Reviewer #2: All comments have been addressed

2. Does this manuscript meet PLOS Global Public Health’s publication criteria? Is the manuscript technically sound, and do the data support the conclusions? The manuscript must describe methodologically and ethically rigorous research with conclusions that are appropriately drawn based on the data presented.

Reviewer #1: Yes

Reviewer #2: Yes

3. Has the statistical analysis been performed appropriately and rigorously?

Reviewer #1: Yes

Reviewer #2: Yes

4. Have the authors made all data underlying the findings in their manuscript fully available (please refer to the Data Availability Statement at the start of the manuscript PDF file)?

Reviewer #1: Yes

Reviewer #2: Yes

5. Is the manuscript presented in an intelligible fashion and written in standard English?

Reviewer #1: Yes

Reviewer #2: Yes

6. Review Comments to the Author

Reviewer #1: Dear Authors,

Thanks for making the efforts to revise the manuscript.

Reviewer #2: All comments have been addressed.

7. PLOS authors have the option to publish the peer review history of their article (what does this mean?). If published, this will include your full peer review and any attached files.

**Do you want your identity to be public for this peer review?** For information about this choice, including consent withdrawal, please see our Privacy Policy.

Reviewer #1: **Yes: **Dr. Kuldeep Singh, ICMR- National Institute of Malaria Research FU Guwahati (Assam)

Reviewer #2: No
